# Outcomes of vaccinations against respiratory diseases in patients with end-stage renal disease undergoing hemodialysis: A systematic review and meta-analysis

**Metalia Puspitasari**[1]☯*, **Prenali D. Sattwika**[1,2]☯, **Dzerlina S. Rahari**[2,3]☯, **Wynne Wijaya**[1]☯, **Auliana R. P. Hidayat**[1]☯, **Nyoman Kertia**[1]‡, **Bambang Purwanto**[4]‡, **Jarir At Thobari**[2,5]‡

1 Department of Internal Medicine, Faculty of Medicine, Public Health and Nursing, Universitas Gadjah Mada/Dr. Sardjito General Hospital, Yogyakarta, Indonesia, 2 Clinical Epidemiology and Biostatistics Unit, Faculty of Medicine, Public Health and Nursing, Universitas Gadjah Mada/Dr. Sardjito General Hospital, Yogyakarta, Indonesia, 3 Department of Epidemiology, Faculty of Medicine, Prince of Songkla University, Hat Yai, Thailand, 4 Department of Internal Medicine, Faculty of Medicine, Universitas Sebelas Maret, Surakarta, Indonesia, 5 Department of Pharmacology and Therapy, Faculty of Medicine, Public Health and Nursing, Universitas Gadjah Mada, Yogyakarta, Indonesia

☯ These authors contributed equally to this work.
‡ NK, BP and JAT also contributed equally to this work.
* metaliapuspitasari@ugm.ac.id

**Data Availability Statement:** Data of extraction table is accessible on Open Science Framework

## Abstract

Due to the nature of the disease, end-stage renal disease (ESRD) patients suffer from dysfunction of the adaptive immune system, which leads to a poorer response to vaccination. Accordingly, it is crucial to evaluate the efficacy and safety of management strategies, including vaccinations, which could potentially reduce the risk of respiratory diseases, such as pneumonia, influenza, or COVID-19, and its associated outcomes. We searched PubMed, CENTRAL, ScienceDirect, Scopus, ProQuest, and Google Scholar databases using designated MeSH keywords. The risk of bias was assessed using ROBINS-I. The quality of evidence was assessed using the GRADE (Grading of Recommendations, Assessment, Development, and Evaluation) approach. Relative risk (RR) and 95% confidence interval (CI) were calculated. Heterogeneity was investigated using forest plots and $I^2$ statistics. This systematic review included a total of 48 studies, with 13 studies of influenza (H1N1 and H3N2) vaccination and 35 studies of COVID-19 vaccination. H1N1 vaccination in ESRD patients undergoing hemodialysis induced lower seroconversion rates (RR 0.62, 95% CI: 0.56–0.68, $p$ <0.00001) and lower seroprotection rates (RR 0.76, 95% CI: 0.70–0.83, $p$ <0.00001) compared to controls. H3N2 vaccination in ESRD patients undergoing hemodialysis yielded lower seroconversion rates (RR 0.76, 95% CI: 0.68–0.85, $p$ <0.00001) and lower seroprotection rates (RR 0.84, 95% CI: 0.77–0.90, $p$ <0.00001) compared to controls. Twenty-nine studies demonstrate significantly lower antibody levels in ESRD patients undergoing hemodialysis compared to the controls following COVID-19 vaccination. This review presents evidence of lower seroconversion and seroprotection rates after vaccination against viral respiratory diseases in patients with ESRD undergoing hemodialysis. Since hemodialysis patients are more susceptible to infection and severe disease

(OSF) portal through this link: https://osf.io/es2ma/
.

**Funding:** The author(s) received no specific funding for this work.

**Competing interests:** The authors have declared that no competing interests exist.

progression, a weakened yet substantial serological response can be considered adequate to recommend vaccination against respiratory diseases in this population. Vaccination dose, schedule, or strategy adjustments should be considered in stable ESRD patients on maintenance hemodialysis.

**Trial registration: Systematic review registration:** https://www.crd.york.ac.uk/prospero/display_record.php?ID=CRD42021255983, identifier: CRD42021255983.

## 1. Introduction

According to the International Society of Nephrology's (ISN) 2019 Global Kidney Health Atlas (GKHA), from 79 countries worldwide, the average number of new end-stage renal disease (ESRD) diagnoses was 144 individuals per million general population. In this population, hemodialysis is the most common technique of predominant renal replacement therapy (RRT) [1]. ESRD patients requiring dialysis are identified as high-risk patients for the severe form of respiratory infections, including pneumonia, influenza, and coronavirus disease 2019 (COVID-19), due to their frequent contact with health care providers and other patients, high burden of comorbid conditions, and altered immune responses [2–5]. Approximately 20% of infections in ESRD patients are attributable to pulmonary causes. The mortality rate of respiratory infections in dialysis patients is 14 to 16-fold higher than in the general population [6].

The high incidence, morbidity, and mortality rate of respiratory infections in ESRD patients have rendered vaccination a vital measure to prevent life-threatening complications. However, ESRD patients mount lower responses to vaccination than healthy individuals due to dysfunction of the adaptive immune system [5, 7, 8]. Furthermore, end-stage renal disease patients have been largely excluded from vaccine trials for safety reasons. Therefore, more convincing evidence regarding the efficacy and safety of vaccinations against respiratory infections is required. This systematic review and meta-analysis aimed to evaluate and summarize the available evidence on the efficacy and safety of vaccination against respiratory infections in ESRD patients undergoing hemodialysis and its associated outcomes to help guide clinical practice and vaccination recommendations.

## 2. Materials and methods

### 2.1 Protocol registration

The protocol of this systematic review has been registered and accepted in PROSPERO with the registration number CRD42021255983 available at https://www.crd.york.ac.uk/prospero/display_record.php?ID=CRD42021255983 (S1 Protocol).

### 2.2 Search strategy and eligibility criteria

We searched PubMed, The Cochrane Central Register of Controlled Trials (CENTRAL), ScienceDirect, Scopus, ProQuest, and Google Scholar for interventional (non-randomized or randomized controlled trials [RCTs]) and observational studies from inception until 20 October 2022. Electronic searches were complemented by manually searching all reference lists of identified studies and reviews for additional studies. We used the MeSH-related keywords such as "end-stage renal disease" AND "hemodialysis" AND ("pneumococcal vaccines" OR "influenza vaccines" OR "COVID-19 vaccines"), as well as their common synonyms. Restrictions involved non-English language and animal studies. The complete search strategy is shown in S1 Appendix.

One reviewer conducted the initial searches. After removing duplicates, three reviewers first scanned all remaining articles by title and abstract. Then, two independent reviewers read the full text of potentially eligible items and decided on which studies to include. Discrepancies were resolved by discussion.

Studies had to meet the following inclusion criteria: (i) original report on the efficacy and safety within six weeks after vaccination against respiratory diseases (pneumococcal, influenza, and COVID-19 vaccines) in adult patients with ESRD undergoing hemodialysis, and (ii) control participants had to be clinically healthy populations who received vaccination against respiratory diseases. We excluded studies in which participants with ESRD in the intervention arm underwent peritoneal dialysis or renal transplant.

### 2.3 Data extraction

Four authors performed data extraction independently using a standardized data extraction form [9]. The following information was extracted from eligible studies: first author, year of publication, study registration, setting, study design, inclusion and exclusion criteria, participant numbers and characteristics, vaccine type, dose, timing and route of administration, outcome definition, and outcome proportion in each arm for dichotomous data or mean and standard deviation (SD) for continuous data. Preferred Reporting Items for Systematic Reviews and Meta-Analyses (PRISMA) guidelines were applied to the search strategy (S1 Checklist) [10]. The complete data extraction table is accessible on the Open Science Framework (OSF) portal via this link: https://osf.io/es2ma/?view_only=87b0e57246704617aa094219a60ba73b.

### 2.4 Risk of bias and quality of evidence assessment

The risk of bias was independently assessed by two authors using a tool for assessing the risk of bias in non-randomized studies of interventions (ROBINS-I) [11]. The tool views each study as an attempt to emulate a hypothetical pragmatic randomized trial and covers seven distinct domains through which bias might be introduced. The judgments within each domain are carried forward to an overall risk of bias judgment across domains for the assessed outcome. The categories for risk of bias judgments are "Low risk", "Moderate risk", "Serious risk", and "Critical risk" of bias. The "No information" category should be used only when insufficient data are reported to permit a judgment. Discrepancies were resolved by discussion. Funnel plots were constructed to check for publication bias in studies included in meta-analyses.

The quality (certainty) of evidence was assessed using the GRADE (Grading of Recommendations, Assessment, Development, and Evaluations) framework. The quality of the overall evidence was rated as one of four levels: very low, low, moderate, and high, based on the assessment of the domains for risk of bias, imprecision, inconsistency, indirectness, and publication bias [12].

### 2.5 Data synthesis and statistical analysis

We examined dichotomous outcomes and expressed results as risk ratio (RR) with a 95% confidence interval (CI). From the included studies, we used the data of seroconversion rate, seroprotection rate, and adverse events rate for meta-analysis. Whenever available, we extracted the data of antibody titer. The analysis was separated between each type of vaccine group. Statistical analysis and generation of forest plots were conducted using Review Manager (RevMan) 5.4 software, with $p<0.05$ deemed statistically significant.

The variability across studies due to heterogeneity was investigated using forest plots and $I^2$ statistics, with $I^2$ values of 0% to 40%, 30% to 60%, 50% to 90%, and 75% to 100%

corresponding to not important, moderate, substantial and considerable levels of heterogeneity, respectively [9].

## 3. Results

### 3.1 Characteristics of included studies

During the initial search, we identified 1080 records from electronic databases, 58 records from Google Scholar, and four additional records from manual searching. After further screening, we included 48 eligible studies (Fig 1). The included studies mostly have cohort design [13–53], five studies are of case-control design [30, 54–57], two studies are of cross-sectional design [58, 59], and 1 study is an open-label clinical trial [60].

Among thirteen studies, eleven provide data on the H1N1 influenza vaccine [15, 21, 30, 32, 42, 49, 50, 55, 57, 58, 60], and 11 studies on the H3N2 influenza vaccine [15, 21, 26, 42, 49, 50,

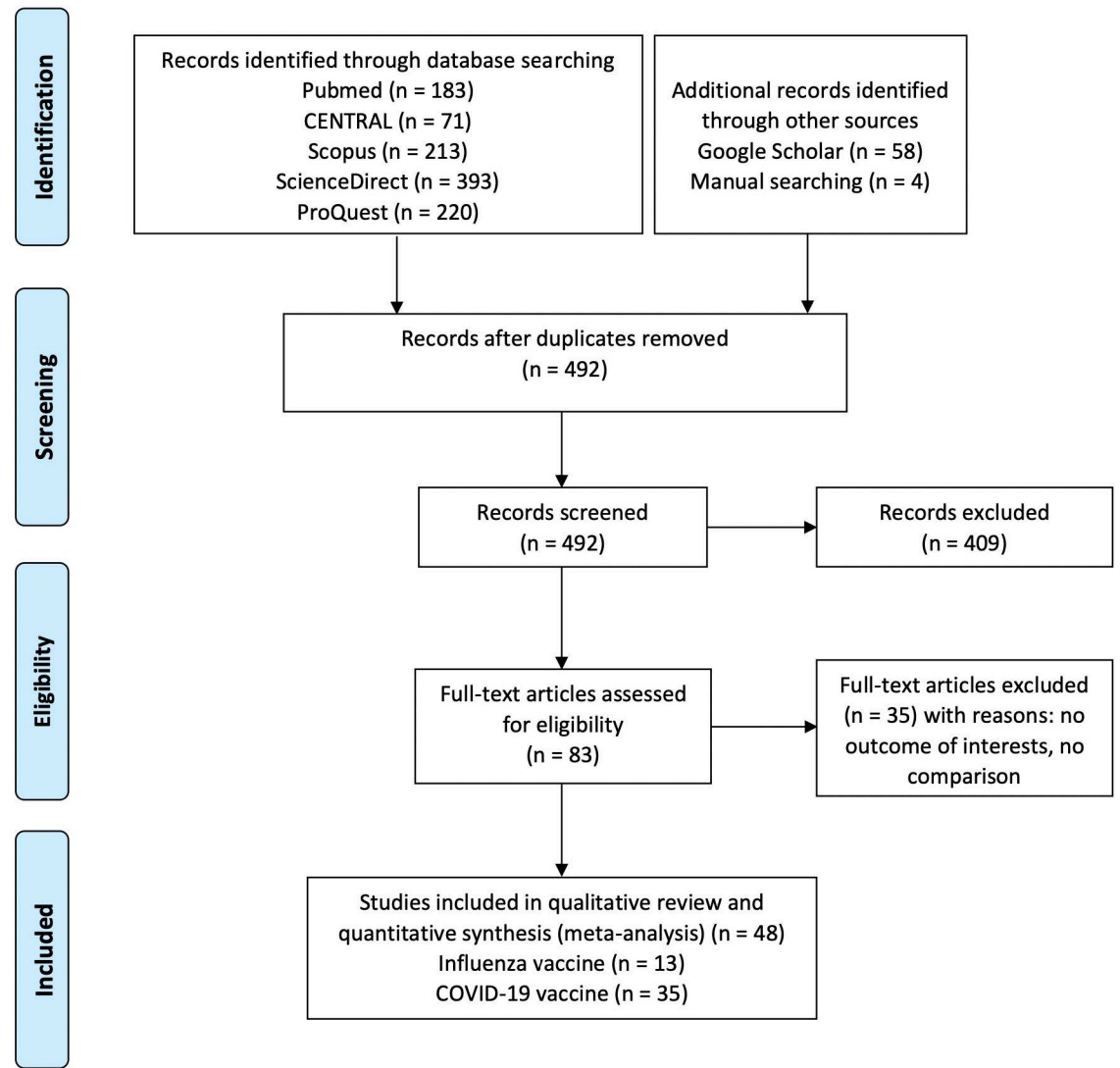

**Fig 1. PRISMA flow diagram for included studies [10].**

55–58, 60]. Most of them used hemagglutination-inhibiting (HI) assay to measure seroconversion and seroprotection. Meanwhile, thirty-five studies on COVID-19 vaccines [13, 16–20, 24, 25, 27–29, 31, 33–41, 43–48, 51–54, 59] used various vaccine platforms (including mRNA-based, inactivated, viral vector, and heterologous vaccines) and various units of measurement for IgG levels and neutralizing antibodies (NAbs) percentage (%) of inhibition. We could not find any studies on pneumococcal vaccines. The outcome of seroconversion and seroprotection rates were assessed for all studies. Table 1 summarizes the characteristics of included studies.

## 3.2 Risk of bias assessment

The risk of bias assessment in each individual study is summarized in Fig 2. We rated the overall risk of bias on the outcome of seroconversion and seroprotection rates to be high risk of bias in two studies and unclear risk of bias in six of thirteen observational studies investigating influenza vaccinations in patients with ESRD undergoing hemodialysis. These risks of bias arise from each domain. Two studies by Versluis in 1985 and 1988 [50, 57] was considered to have high risks of bias due to selection bias in sequence generation and selective reporting (reporting bias). From thirty-five included observational studies of COVID-19 vaccinations for patients with ESRD undergoing hemodialysis, twenty-one studies showed an unclear risk of bias due to bias in sequence generation (selection bias) [13, 16, 22–24, 34, 40, 41, 48, 52], blinding of participants and personnel (performance bias) [29, 35, 36, 41, 47], blinding of outcome assessment (performance bias) [33], incomplete outcome data (attrition bias) [13, 20, 54], or selective reporting (reporting bias) [17, 51, 59]. Funnel plots to assess publication bias in studies included for meta-analyses were also constructed and displayed in S2 Appendix.

## 3.3 Outcome

This section discusses the outcomes of vaccination in patients with ESRD undergoing hemodialysis consisting of efficacy and adverse events outcomes in included studies.

**3.3.1 H1N1 vaccine.** For the H1N1 vaccine, vaccination in ESRD patients undergoing hemodialysis showed lower seroconversion and seroprotection rates compared to controls. Ten of the included studies reported the outcome of seroconversion rate. H1N1 vaccination in patients with ESRD undergoing hemodialysis induced lower seroconversion rates (Fig 3A, with 10 studies, 1191 participants: RR 0.62, 95% CI: 0.56–0.68, $p < 0.00001$) with substantial heterogeneity ($I^2 = 81\%$). One study by Labriola in 2011 utilized a seroneutralization assay to measure antibody level and reported a significantly lower seroconversion rate in HD patients (64,2%) compared to controls (93,8%) (p = 0.002) [30]. Seroprotection rate was lower in ESRD patients receiving H1N1 vaccines compared to controls (Fig 3B, with 7 studies, 1001 participants: RR 0.76, 95% CI: 0.70–0.83, p<0.00001) with considerable heterogeneity ($I^2 = 96\%$). There was only one study reporting adverse events following vaccinations of H1N1 with 2 of 53 patients with ESRD experiencing moderate local pain at the site of injection with no adverse events observed in the control group [30].

**3.3.2 H3N2 vaccine.** H3N2 vaccination in patients with ESRD undergoing hemodialysis produced lower rates of seroconversion compared to controls (Fig 4A, with 10 studies, 1012 participants: RR 0.76, 95% CI: 0.68–0.85, $p < 0.00001$) with moderate heterogeneity ($I^2 = 43\%$) and lower rates of seroprotection (Fig 4B, with 6 studies, 754 participants: RR 0.84, 95% CI: 0.77–0.90, $p < 0.00001$) with considerable heterogeneity ($I^2 = 85\%$). A study by Nikoskelainen in 1982 determined the antibody responses with single radial hemolysis (SRH) technique and demonstrated a higher seroconversion rate in HD patients (92%) compared to controls (88%) [56]. In terms of adverse events, ESRD patients undergoing hemodialysis experienced lower

**Table 1. Characteristics of included studies.**

| No | Author, year | Setting | Study design | Study population | Number of participants | Comparison | Vaccination | Outcomes | Additional Remarks |
|---|---|---|---|---|---|---|---|---|---|
| **Influenza Vaccines** | | | | | | | | | |
| 1 | Antonen 2003 | Finland | Cohort | Hemodialysis patients | Exposure: 23 Comparison: 26 | Military conscript | Influenza vaccine (H3N2) | Seroprotection | Platform of vaccine: inactivated trivalent vaccine Method to measure antibody response (unit): haemagglutination-inhibiting (HI) antibodies (%), pre and 5 weeks after vaccination |
| 2 | Beyer 1987 | Netherlands | Cohort | Hemodialysis patients | H3N2 Exposure: 73 Comparison: 20 H1N1 Exposure: 91 Comparison: 25 | Healthy controls | Influenza vaccine (H3N2, H1N1) | Seroconversion, seroprotection | Platform of vaccine: inactivated trivalent vaccine Method to measure antibody response (unit): haemagglutination-inhibiting (HI) antibodies (%), pre and 4 weeks after vaccination |
| 3 | Eiselt 2016 | Czech Republic | Cohort | Hemodialysis patients | H3N2 Exposure: 133 Comparison: 40 H1N1 Exposure: 133 Comparison: 40 | Healthy Controls | Influenza vaccine (H3N2, H1N1) | Seroconversion, seroprotection | Platform of vaccine: inactivated trivalent vaccine Method to measure antibody response (unit): haemagglutination-inhibiting (HI) antibodies (%), pre and 4 weeks after vaccination |
| 4 | Hodges 1979 | USA | Cohort | Hemodialysis patients | Exposure: 13 Comparison: 41 | Healthy controls | Influenza vaccine (H3N2) | Seroconversion | Platform of vaccine: inactivated bivalent split-virus Method to measure antibody response (unit): haemagglutination-inhibiting (HI) antibodies (%), before and after vaccination |
| 5 | Krairittichai 2013 | Thailand | Cross-sectional | Hemodialysis patients | H3N2 Exposure: 22 Comparison: 6 H1N1 Exposure: 23 Comparison: 20 | Healthy controls | Influenza vaccine (H3N2, H1N1) | Seroconversion, seroprotection | Platform of vaccine: inactivated trivalent vaccine Method to measure antibody response (unit): haemagglutination-inhibiting (HI) antibodies (%), before and 6 weeks after vaccination |
| 6 | Labriola 2011 | Belgium | Case-control | Hemodialysis patients | Exposure: 53 Comparison: 32 | Healthy controls | Influenza vaccine (H1N1) | Seroconversion, adverse effects | Platform of vaccine: monovalent adjuvanted influenza A/California/ 2009 (H1N1) vaccine Method to measure antibody response (unit): seroneutralization (SN) assay (%) day 0 and 30 |

*(Continued)*

**Table 1.** (Continued)

| No | Author, year | Setting | Study design | Study population | Number of participants | Comparison | Vaccination | Outcomes | Additional Remarks |
|---|---|---|---|---|---|---|---|---|---|
| 7 | Lertdumrongluk 2011 | Thailand | Cohort | Hemodialysis patients | Exposure: 44 Comparison: 149 | Healthy controls | Influenza H1N1 vaccine | HI antibody titer, seroconversion | Platform of vaccine: a single dose of non-adjuvanted 2009 influenza A (H1N1) vaccine (Paneza®) Method to measure antibody response (unit): Hemagglutination inhibition (HI) assays (GMT), before, 4 weeks, and 24 weeks after vaccination |
| 8 | Mastalerz-Migas 2015 | Poland | Case-control | Hemodialysis patients | H3N2 Exposure: 71 Comparison: 63 H1N1 Exposure: 71 Comparison: 63 | Healthy controls | Influenza vaccine (H3N2, H1N1) | Seroconversion, seroprotection | Platform of vaccine: inactivated trivalent vaccine Method to measure antibody response (unit): haemagglutination-inhibiting (HI) antibodies (%), before and after vaccination |
| 9 | Nikoskelainen 1982 | Finlandia | Case-control | Hemodialysis patients | Exposure: 12 Comparison: 40 | Healthy controls | Influenza vaccine (H3N2) | Seroconversion | Platform of vaccine: inactivated trivalent vaccine Method to measure antibody response (unit): single radial hemolysis (SRH) technique |
| 10 | Song 2006 | South Korea | Cohort | Hemodialysis patients | Exposure: 50 Comparison: 50 | Healthy controls | Influenza vaccine (H3N2, H1N1) | HI antibody titer, seroresponse, seroprotection | Platform of vaccine: a single dose of trivalent inactivated split vaccine (Inflexin®) (H1N1, H3N2, B/Hongkong) Method to measure antibody response (unit): hemagglutination-inhibiting (HI) antibodies (%), 4 weeks after vaccination |
| 11 | Versluis 1985 | Netherlands | Cohort | Hemodialysis patients | H3N2 Exposure: 10 Comparison: 4 H1N1 Exposure: 10 Comparison: 6 | Healthy controls | Influenza vaccine (H3N2, H1N1) | Seroconversion | Platform of vaccine: whole virus vaccine Method to measure antibody response (unit): haemagglutination-inhibiting (HI) antibodies (%) at day 0, 30, and 60 |
| 12 | Versluis 1988 | Netherlands | Case-control | Hemodialysis patients | H3N2 Exposure: 101 Comparison: 30 H1N1 Exposure: 101 Comparison: 30 | Healthy controls | Influenza vaccine (H3N2, H1N1) | Seroconversion | Platform of vaccine: inactivated trivalent vaccine Method to measure antibody response (unit): haemagglutination-inhibiting (HI) antibodies (%), pre and 4 weeks after vaccination |

(*Continued*)

**Table 1.** (Continued)

| No | Author, year | Setting | Study design | Study population | Number of participants | Comparison | Vaccination | Outcomes | Additional Remarks |
|---|---|---|---|---|---|---|---|---|---|
| 13 | Vogtlander 2004 | Netherlands | Cohort | Hemodialysis patients | Exposure: 44 Comparison: 19 | Hospital staff | Influenza vaccine (H3N2, H1N1) | HI antibody titer, seroconversion, seroprotection | Platform of vaccine: Method to measure antibody response (unit): SARS-CoV-2 IgG II Quant assay (AU/mL), 5 weeks after second dose |
| **COVID-19 Vaccines** | | | | | | | | | |
| 14 | Ahmed 2022 | Egypt | Cohort | Hemodialysis patients | Exposure: 44 Comparison: 22 | Non-renal patients | Inactivated or mRNA SARS-CoV-2 vaccines | IgG level and adverse events | Platform of vaccine: Sinopharm Method to measure antibody response (unit): SARS-CoV-2 IgG ELISA assay (AU/ml) at 30 days after second dose |
| 15 | Bai 2022 | Pakistan | Cross-sectional | Hemodialysis patients | Exposure: 50 Comparison: 31 | Healthy individuals | Inactivated or mRNA SARS-CoV-2 vaccines | IgG level | Platform of vaccine: BBIBP-CorV produced by Sinopharm Beijing or CoronaVac® Method to measure antibody response (unit): Cobas® Elecsys Anti-SARS-CoV-2 S Immunoassay (Roche Diagnostics, Basel, Switzerland) (U/ml), at baseline, 20 days after the first dose, and 3 weeks after the second dose |
| 16 | Boongird 2021a | Thailand | Cohort | Hemodialysis patients | Exposure: 60 Comparison: 30 | Healthy controls | CoronaVac vaccine | IgG level, seroconversion | Platform of vaccine: two doses of CoronaVac vaccine Method to measure antibody response (unit): semiquantitative SARS-CoV-2 IgG assay (Abbott Diagnostics) at 2 weeks after second dose |
| 17 | Boongird 2022b | Thailand | Cohort | Hemodialysis patients | Exposure: 31 Comparison: 30 | Healthy control | Inactivated whole-virus SARS-CoV-2 vaccine | IgG levels, NAbs % inhibition | Platform of vaccine: two doses of CoronaVac® Method to measure antibody response (unit): SARS-CoV-2 IgG II Quant; Abbott Diagnostics (AU/ml) and sVNT (Euroimmun kits), at baseline, 4 weeks after the first dose, and 2 weeks after the second dose |

(*Continued*)

**Table 1.** (*Continued*)

| No | Author, year | Setting | Study design | Study population | Number of participants | Comparison | Vaccination | Outcomes | Additional Remarks |
|---|---|---|---|---|---|---|---|---|---|
| 18 | Bruminhent 2022 | Thailand | Cohort | Hemodialysis patients | Exposure: 31 Comparison: 16 | Healthy controls | CoronaVac vaccine | IgG level, NAbs % inhibition, Seroconversion | Platform of vaccine: two doses of CoronaVac vaccine Method to measure antibody response (unit): Abbott SARS-CoV-2 IgG II Quantification assay (Abbott Diagnostics, USA) (BAU/mL) and SARS-CoV-2 NeutraLISA surrogate neutralization assay (Euroimmun) (%) at 2 weeks after second dose |
| 19 | Danthu 2021 | France | Cohort | Hemodialysis patients | Exposure: 78 Comparison: 7 | Healthy controls | Pfizer BNT162b2 vaccine | IgG level, seroconversion | Platform of vaccine: two doses of CoronaVac vaccine Method to measure antibody response (unit): the LIAISON SARS-CoV-2 TrimericS IgG (DiaSorin, Saluggia, Italy) (AU/mL) and Abbott Alinity SARS-CoV-2 IgG, Chicago, IL, USA (%) at 0, 14, 28, 36, and 58 days after the first dose (8 days after second dose) |
| 20 | Dheir 2022 | Turkey | Cohort | Hemodialysis patient | Exposure: 50 Comparison: 41 | Healthy group | CoronaVac vaccine | IgG level | Platform of vaccine: two doses of inactivated vaccine CoronaVac Method to measure antibody response (unit): SARS-CoV-2 IgG II Quant; Abbott Diagnostics (AU/ml) at 28 days, 3 and 6 months |
| 21 | Fu 2022 | Taiwan | Cohort | Hemodialysis patients | Exposure: 385 Comparison: 66 | Healthcare workers | ChAdOx1 nCoV-19 vaccines | IgG level, seroconversion | Platform of vaccine: two doses of ChAdOx1 nCoV-19 vaccines Method to measure antibody response (unit): Elecsys® Anti-SARS-CoV-2-S immunoassay (U/mL), 4 weeks after second dose |
| 22 | Fucci 2022 | Italy | Cohort | Hemodialysis patients | Exposure: 155 Comparison: 77 | Healthy control | COVID-19 mRNA vaccination | IgG level, Seroconversion | Platform of vaccine: two doses of BNT162b2 vaccines Method to measure antibody response (unit): COVID-19 QuantiGEM SARS-CoV-2 IgG ELISA Kit CE-IVD (ng/mL), 33–45 days after the first dose (12–24 days after the second dose) |

(*Continued*)

**Table 1.** (Continued)

| No | Author, year | Setting | Study design | Study population | Number of participants | Comparison | Vaccination | Outcomes | Additional Remarks |
|---|---|---|---|---|---|---|---|---|---|
| 23 | Grupper 2021 | Israel | Cohort | Hemodialysis patients | Exposure: 56 Comparison: 95 | Health care workers | Pfizer BNT162b2 vaccine | IgG level | Platform of vaccine: BNT162B2 Method to measure antibody response (unit): a chemiluminescent microparticle immunoassay (SARS-CoV-2 IgG II Quant assay on an ARCHITECT analyzer; Abbott) (AU/ml) 4 weeks after second dose |
| 24 | Haase 2022 | Germany | Cohort | Hemodialysis patients | Exposure: 137 Comparison: 24 | Immunocompetent medical personnel | ChAdOx1-S-nCoV-19 and BNT162B2 | IgG level | Platform of vaccine: ChAdOx1-S-nCoV-19 and BNT162B2 Method to measure antibody response (unit): The SARS-CoV-2-IgG-II-Quant-assay is an automated CMIA (BAU/ml) 6 weeks after second dose |
| 25 | Jahn 2021 | Germany | Cohort | Hemodialysis patients | Exposure: 72 Comparison: 16 | Healthcare workers | Pfizer BNT162b2 vaccine | IgG level, Seroconversion rate | Platform of vaccine: two doses of mRNA-based BNT162b2 vaccines Method to measure antibody response (unit): anti-SARS-CoV-2 IgG CLIA LIAISON® SARS-CoV-2 TrimericS IgG assay (AU/ml), two weeks after second dose |
| 26 | Kim 2022 | South Korea | Cohort | Hemodialysis patients | Exposure: 100 Comparison: 100 | Hospital workers | HD: ChAdOx1/ BNT162b2 Control: ChAdOx1/ ChAdOx1 | IgG level, seroconversion | Platform of vaccine: two doses of SARS-CoV-2 vaccines (ChAdOx1/ BNT162b2) Method to measure antibody response (unit): ARCHITECT IgG II Quant test (Abbott Laboratories) (AU/ml), two months after second dose |
| 27 | Kolb 2021 | Germany | Cohort | Hemodialysis patients | Exposure: 32 Comparison: 78 | Healthy control | BNT162b2 or mRNA-1273 vaccine | IgG level, seroconversion | Platform of vaccine: two doses of mRNA-based SARS-CoV-2 vaccines (BNT162b2 or mRNA-1273) Method to measure antibody response (unit): Anti-SARS-CoV-2 QuantiVac ELISA (Euroimmun) (BAU/ml), 14 days after second dose |

*(Continued)*

**Table 1.** (*Continued*)

| No | Author, year | Setting | Study design | Study population | Number of participants | Comparison | Vaccination | Outcomes | Additional Remarks |
|---|---|---|---|---|---|---|---|---|---|
| 28 | Labriola 2021 | Belgium | Cohort | Hemodialysis patients | Exposure: 24 Comparison: 33 | Non-dialyzed nursing home resident | BNT162b2 | IgG level, seroconversion | Platform of vaccine: two doses of BNT162b2 vaccines Method to measure antibody response (unit): electrochemiluminescent assays from Elecsys (U/ml), 28 days after first dose (7 days after second dose) |
| 29 | Lesny 2021 | Germany | Cohort | Hemodialysis patient | Exposure: 23 Comparison: 18 | Hemodialysis patient with prior COVID-19 infection | First mRNA- or vector-based SARS-CoV-2 vaccination | IgG level | Platform of vaccine: first mRNA- or vector-based SARS-CoV-2 vaccination Method to measure antibody response (unit): The SARS-CoV-2 IgG II Quant assay is an automated CMIA (AU/ml) 2 weeks after first dose |
| 30 | Matsunami 2021 | Japan | Cohort | Hemodialysis patients | Exposure: 78 Comparison: 38 | Healthy controls | Pfizer BNT162b2 vaccine | IgG level | Platform of vaccine: BNT162B2 Method to measure antibody response (unit): system Elecsys® Anti-SARS-CoV-2 S RUO (Roche Diagnostics, Basel, Switzer-land) (U/ml) 2–8 weeks after second dose |
| 31 | Murt 2021 | Turkey | Cohort | Hemodialysis patients | Exposure: 85 Comparison: 103 | Healthy controls | inactivated or mRNA SARS-CoV-2 vaccines | IgG level | Platform of vaccine: CoronaVac® or BNT162b2 Method to measure antibody response (unit): Abbott SARS-CoV-2 IgG II Quant (Chicago, USA) (AU/ml), 21–28 days after the second dose |
| 32 | Panizo 2022 | Spain | Cohort | Hemodialysis patients | Exposure: 52 Comparison: 18 | Healthy control | mRNA-1273 or BNT162b2 vaccine | IgG level, seroconversion | Platform of vaccine: two doses of mRNA vaccines (mRNA-1273 or BNT162b2) Method to measure antibody response (unit): Roche Elecsys® Anti-SARS-CoV-2 S (U/ml), 15 days and 3 months after second dose |
| 33 | Park 2022 | South Korea | Cohort | Hemodialysis patients | Exposure: 33 Comparison: 55 | Healthy controls | ChAdOx1/ ChAdOx1 or ChAdOx1/ BNT162b2 (for HD patients) | IgG level, NAbs % inhibition, seroconversion | Platform of vaccine: two doses of ChAdOx1 or mix-and-match ChAdOx1/ BNT162b2 (only for HD patients) Method to measure antibody response (unit): Roche Elecsys® Anti-SARS-CoV-2 S (U/ml) and cPass™ SARS-CoV-2 Neutralization Antibody Detection Kit, 56 days after first dose (28 days after second dose) |

(*Continued*)

**Table 1.** (Continued)

| No | Author, year | Setting | Study design | Study population | Number of participants | Comparison | Vaccination | Outcomes | Additional Remarks |
|---|---|---|---|---|---|---|---|---|---|
| 34 | Piotrowska 2022 | Poland | Cohort | Hemodialysis patients | Exposure: 35 Comparison: 34 | Healthy controls | Pfizer BNT162b2 vaccine | Anti-S IgG level, seroconversion rate | Platform of vaccine: two doses of BNT162b2 vaccines Method to measure antibody response (unit): DiaSorin LIAISON®SARS-CoV-2 S1/S2 IgG (AU/ml), 21 days after the first dose and 14–21 days after the second dose |
| 35 | Piscitani 2022 | Italy | Case-control | Hemodialysis patients | Exposure: 21 Comparison: 16 | Healthy controls | Pfizer BNT162b2 vaccine | IgG level | Platform of vaccine: BNT162b2 Method to measure antibody response (unit): fluorescence polarization immunoassay (FPIA) (Roche®) (IU/ml), after second dose |
| 36 | Scharpe 2009 | Belgium | Open-label study | Hemodialysis patients | H1N1 Exposure: 201 Comparison: 41 H1N1 Exposure: 201 Comparison: 41 | Healthy controls | Influenza vaccine (H3N2, H1N1) | Seroprotection, seroconversion, adverse event | Platform of vaccine: inactivated trivalent vaccine Method to measure antibody response (unit): haemagglutination-inhibiting (HI) antibodies (%), before and 1 month after vaccination |
| 37 | Schrezenmeier 2021 | Germany | Cohort | Hemodialysis patients | Exposure: 36 Comparison: 44 | Healthy controls | Tozinameran (BNT162b2 BioNTech/ Pfizer) | Seroconversion, Anti-SARS-CoV-2 antibody titers | Platform of vaccine: BNT162b2 BioNTech/ Pfizer Method to measure antibody response (unit): anti-SARSCoV-2-S1 IgG/IgA ELISA (Euroimmun, Lübeck, Germany) (IU/ml), week 1 and week 3–4 |
| 38 | Simon 2021 | Austria | Cohort | Hemodialysis patients | Exposure: 81 Comparison: 80 | Healthy controls | COVID-19 mRNA vaccination | Anti-SARS-CoV-2 antibody titers, adverse event | Platform of vaccine: mRNA vaccine BNT162b2 Method to measure antibody response (unit): Elecsys® Anti-SARS-CoV-2 test (U/ml), 21 days after second dose |
| 39 | Smith 2022 | United Kingdom | Cohort | Hemodialysis patients | Exposure: 260 Comparison: 144 | Healthy controls | ChAdOx1 BNT162b2 | IgG level, seroconversion | Platform of vaccine: mRNA vaccine BNT162b2 Method to measure antibody response (unit): Elecsys® Anti-SARS-CoV-2 test (MFI titer), 4–6 weeks after complete vaccination |

(*Continued*)

**Table 1.** (Continued)

| No | Author, year | Setting | Study design | Study population | Number of participants | Comparison | Vaccination | Outcomes | Additional Remarks |
|---|---|---|---|---|---|---|---|---|---|
| 40 | Speer 2021a | Germany | Cohort | Hemodialysis patients | Exposure: 124 Comparison: 20 | Healthy controls | BNT162b2 | Anti-S1 IgG level, NAbs % inhibition, seroconversion | Platform of vaccine: two doses of BNT162b2 vaccines. Method to measure antibody response (unit): SARS-CoV-2 Total Assay (Siemens) (semiquantitative index) and SARS-CoV-2 surrogate virus neutralizing assay (Medac) (%), at 20 (18–23) days for HD and 19 (19–23) days for control after second dose |
| 41 | Speer 2021b | Germany | Cohort | Hemodialysis patients | Exposure: 22 Comparison: 46 | Healthy controls | BNT162b2 | Anti-S1 IgG level, NAbs % inhibition, seroconversion | Platform of vaccine: two doses of BNT162b2 vaccines. Method to measure antibody response (unit): SARS-CoV-2 Total Assay (Siemens) (semiquantitative index) and SARS-CoV-2 surrogate virus neutralizing assay (Medac) (%), 20 days after second dose |
| 42 | Speer 2021c | Germany | Cohort | Hemodialysis patients | Exposure: 30 Comparison: 18 | Healthy controls | BNT162b2 | Anti-S1 IgG level, NAbs % inhibition, seroconversion | Platform of vaccine: two doses of BNT162b2 vaccines. Method to measure antibody response (unit): SARS-CoV-2 Total Assay (Siemens) (semiquantitative index) and SARS-CoV-2 surrogate virus neutralizing assay (Medac) (%), 21 days after second dose |
| 43 | Strengert 2021 | Germany | Cohort | Hemodialysis patients | Exposure: 81 Comparison: 34 | Healthcare workers | BNT162b2 | IgG level, NAbs % inhibition, seroconversion | Platform of vaccine: two doses of BNT162b2 vaccines. Method to measure antibody response (unit): multiplex immunoassay MULTICOV-AB (MFI) and anti-SARS-CoV-2-QuantiVac-ELISA IgG (Euroimmun), at 21 days after second dose |

*(Continued)*

**Table 1.** (Continued)

| No | Author, year | Setting | Study design | Study population | Number of participants | Comparison | Vaccination | Outcomes | Additional Remarks |
|---|---|---|---|---|---|---|---|---|---|
| 44 | Tillmann 2021 | Germany | Cohort | Hemodialysis patients | Exposure: 95 Comparison: 60 | Healthy staff | BNT162b2 | Neutralizing antibodies % inhibition, seroconversion | Platform of vaccine: two doses of BNT162b2 or ChAdOx1 vaccines. Method to measure antibody response (unit): GenScript SARS-CoV-2 Surrogate Virus Neutralization Test Kit (%), 4–5 weeks after second dose |
| 45 | Van Praet 2021 | Belgium | Cohort | Hemodialysis patients | Exposure: 543 Comparison: 75 | Healthy individuals | BNT162b2 or mRNA-1273 | IgG level, seroconversion | Platform of vaccine: two doses of BNT162b2 or mRNA-1273 vaccines. Method to measure antibody response (unit): SARS-CoV-2 IgG II Quant assay (AU/mL), 5 weeks after second dose |
| 46 | Wang 2022 | Taiwan | Cohort | Hemodialysis patients | Exposure: 204 Comparison: 34 | Healthcare workers | ChAdOx1 | Anti-RBD IgG level, seroconversion, adverse events | Platform of vaccine: two doses of ChAdOx1 vaccines Method to measure antibody response (unit): Abbott AdviseDx SARS-CoV-2 IgG II assay (AU/mL), T1, four to six weeks after the first dose of vaccine, (efforts were made to try to coordinate with routine blood tests to reduce the negative effects of the extra blood draw); T2, one week before the second dose (to establish baseline concentration); and T3, four to six weeks after the second dose (to assess the antibody response after both injections of the vaccine were complete) |
| 47 | Yau 2021 | Canada | Cohort | Hemodialysis patients | Exposure: 142 Comparison: 35 | Healthcare workers | BNT162b2 | IgG level (anti spike, anti-RBD, anti-NP), seroconversion | Platform of vaccine: two doses of BNT162b2 vaccines Method to measure antibody response (unit): automated enzyme-linked immunosorbent assay platform, baseline and weekly until 14 days after second vaccine dose |

*(Continued)*

**Table 1.** (Continued)

| No | Author, year | Setting | Study design | Study population | Number of participants | Comparison | Vaccination | Outcomes | Additional Remarks |
|----|-------------|---------|--------------|------------------|------------------------|------------|-------------|----------|--------------------|
| 48 | Zhao 2022 | Japan | Cohort | Hemodialysis patients | Exposure: 65 Comparison: 500 | Residents | BNT162b2 | Anti-S1 IgG level, NAbs % inhibition, seroconversion | Platform of vaccine: two doses of BNT162b2 vaccines Method to measure antibody response (unit): the CLIA assay with iFlash 3000 (YHLO Biotech, Shenzhen, China) and iFlash-2019-nCoV series (YHLO Biotech, Shenzhen, China) at 105 days (range 70–112) for dialysis group and 117 days (range 15–170) for control group after second dose |

adverse events rated compared to the control group (HD: 22% vs control: 56%, $p = 0.003$) [60]. ESRD patients developed fewer local symptoms and had fewer symptoms of generalized myalgia and headache.

**3.3.3 COVID-19 vaccine.** Thirty-five studies investigated the antibody responses after COVID-19 vaccination in ESRD patietns undergoing hemodialysis compared to healthy controls. These studies used various vaccine platforms (including mRNA, inactivated, viral vector and heterologous vaccines) as well as different units of measurements. Table 2 summarizes the comparison of IgG levels between HD and control groups following COVID-19 vaccination obtained from the 30 studies [16–20, 22–25, 27–29, 31, 33–36, 38–40, 43, 44, 46–48, 51, 53, 54, 59]. Overall, twenty-nine studies demonstrated lower IgG levels after COVID-19 vaccination in HD patients compared to healthy controls, whereas only one study by Panizo showed a contrary finding [36].

A study by Haase in 2022 reported higher spike IgG levels in HD patients receiving heterologous vaccination with ChAd/BNT (1744 [267–2840] BAU/mL) compared to HD patients receiving homologous vaccination with BNT/BNT (361 [120–936] BAU/mL), ChAd/ChAd (100 [41–346] BAU/mL), and healthy controls (650 [217–1402] BAU/mL). However, the study did not differentiate the spike IgG levels between different vaccine platforms combinations in the control group [25]. Lesny 2021 showed a lower mean IgG level in HD patients (1.6 [0–14.5] AU/mL) compared to controls (73.1 [16.1–1324.5] AU/mL) after only the first dose of vaccination. This study also reported a lower ACE 2 receptor binding inhibition capacity in HD patients (5.0% [3.1–10.4]) compared to healthy controls (10.5% [6.0–40.9]) [33].

ESRD patients undergoing hemodialysis presented with a lower number of adverse events compared to the control group (Fig 5, with 5 studies, 677 participants: RR 0.34, 95% CI: 0.27–0.42, $p < 0.00001$) with substantial heterogeneity ($I^2 = 88\%$) [13, 20, 25, 40, 51].

## 4. Discussion

### 4.1 H1N1 vaccine

In this present study, the intensity of immune response to vaccinations for viral respiratory diseases such as influenza (H1N1 and H3N2) and COVID-19 was inferior in patients with ESRD undergoing hemodialysis compared to healthy subjects. Serological conversion

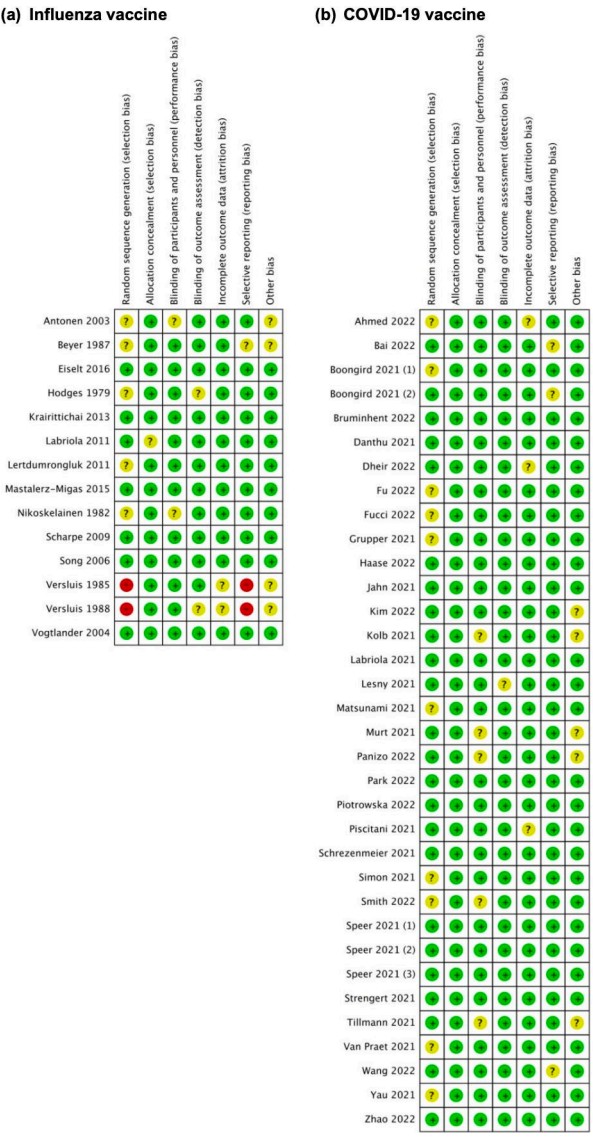

**Fig 2. Assessment risk of bias in non-randomized studies of interventions.** (a) Influenza vaccine and (b) COVID-19 vaccine studies (green: low risk, yellow: moderate risk, red: serious risk, black: critical risk).

following influenza vaccinations was determined as the outcome measure of efficacy due to the unavailability of hemagglutination-inhibiting antibody titers in most included studies. Ten heterogenous studies were used to generate pooled estimates of seroconversion rate after H1N1 vaccination in patients with ESRD receiving hemodialysis and healthy controls. Except for two studies by Versluis in 1985 and Song in 2006, all investigations found a significant reduction in seroconversion rate in patients with ESRD on hemodialysis compared to healthy controls. The pooled estimates showed a 38% decrease in seroconversion rate in patients with ESRD undergoing hemodialysis. This result is consistent with previous literature reviews in which patients with CKD and ESRD experience significant dysregulation in the adaptive immunity, including T cells and B cells, which impairs vaccine response. The B cells changes in patients with CKD/ESRD include a decrease in the number of B cells, B-cell activating factor, B-cell lymphoma 2 (Bcl-2), and an increase in apoptosis. All of these changes result in the

### (a) Seroconversion rate

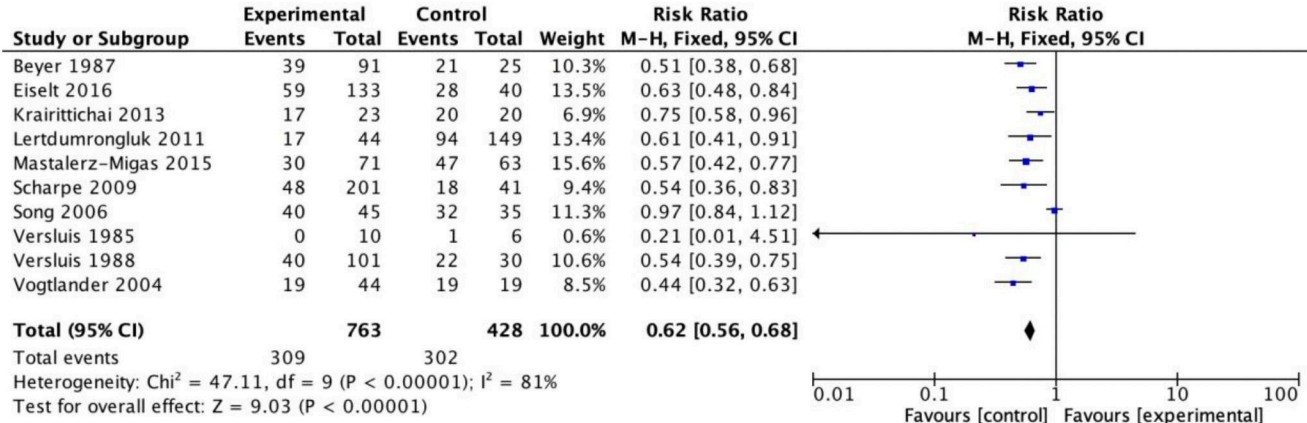

### (b) Seroprotection rate

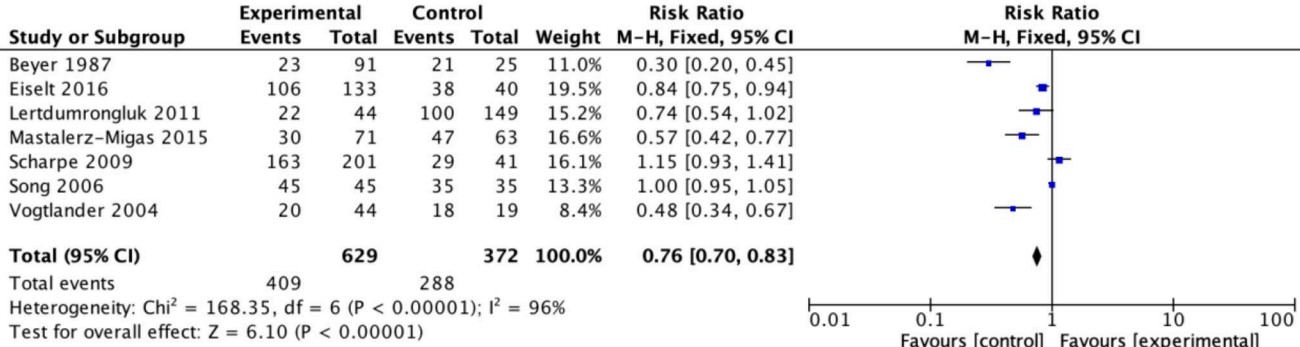

**Fig 3. Forest plot of studies reporting.** (a) seroconversion rate and (b) seroprotection rate after H1N1 vaccination in patients with ESRD undergoing hemodialysis.

depletion of serological response [5]. The insignificant results of Versluis in 1985 could be related to the small sample size [50]. However, the power of this study (weighted at 0.6%) is insufficient to alter the outcome of our analysis. The limitation of the study by Song in 2006 was that a previous vaccination history was not considered and there was a considerable number of dropouts—which might affect the seroconversion rate [42].

Pooled estimates of seroprotection rate after H1N1 vaccination were derived from seven studies with considerable heterogeneity ($I^2 = 96\%$). Five of the seven studies showed a significant reduction in seroprotection rate in patients with ESRD on hemodialysis compared to healthy controls, which is consistent with a previous literature review of adaptive immune dysfunction in patients with CKD/ESRD [5]. The study by Scharpe in 2009 demonstrated an insignificantly higher seroprotection rate in hemodialysis patients compared to healthy controls, both in subjects with and without baseline seroprotection before vaccination. We assume that this is attributable to (1) a higher seroprotection rate in hemodialysis patients due to more frequent immunizations the previous year and (2) the role of recent dialysis procedural improvements and therapeutic drug advancements [60]. However, only further studies with a larger number of patients will be able to confirm or refute this hypothesis. As mentioned before, the study by Song in 2006 had several limitations that might have affected the outcomes [42].

## (a) Seroconversion rate

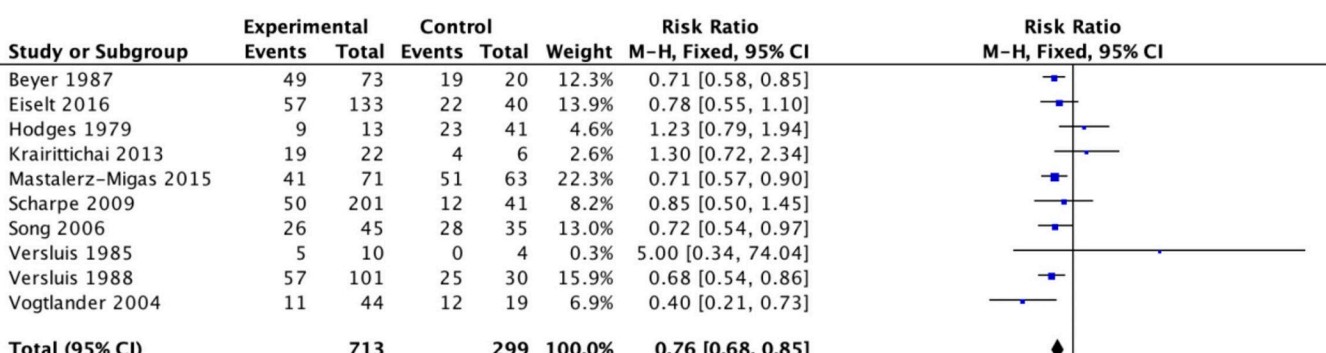

## (b) Seroprotection rate

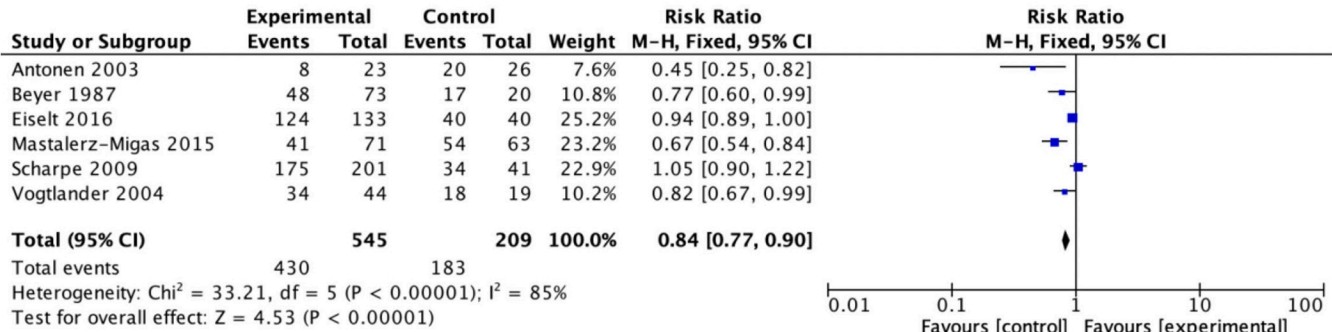

**Fig 4. Forest plot of studies reporting.** (a) seroconversion rate and (b) seroprotection rate (below) after H3N2 vaccination in patients with ESRD undergoing hemodialysis.

We found only one study that measured the adverse events after H1N1 influenza vaccination as an outcome. Labriola in 2011 reported that 2 out of 53 hemodialysis patients presented with moderate local pain at the site of injection. No other side effects associated with the vaccination were observed in hemodialysis patients. However, the number of hemodialysis patients included in the study was small. The results were limited in generalizability due to a larger Caucasian population in the study group. In addition, the intensity and types of local adverse reactions were not characterized [30]. As a result, further studies with larger sample sizes and more diverse subjects are required to evaluate adverse events following H1N1 vaccination in hemodialysis patients.

### 4.2 H3N2 vaccine

Pooled estimates of seroconversion rate after H3N2 vaccination in patients with ESRD undergoing hemodialysis were derived from 10 studies with moderate heterogeneity. Our findings showed a 24% decrease in seroconversion rate in hemodialysis patients, indicating impaired serological response compared to healthy subjects, which is consistent with a recent literature review [5]. In six of the ten studies, the seroconversion rates of hemodialysis patients were shown to be significantly lower than healthy controls.

**Table 2. Comparison of IgG levels between HD and control group after COVID-19 vaccination extracted from 30 studies [16–20, 22–25, 27–29, 31, 33–36, 38–40, 43, 44, 46–48, 51, 53, 54, 59].**

| No | Author | Unit of IgG level | Time to measurement after vaccination | Baseline data | | | | Follow up data | | | |
|----|--------|------|------|------|------|------|------|------|------|------|------|
| | | | | HD group | | Control group | | HD group | | Control group | |
| | | | | N | Mean (SD) or median (IQR) | N | Mean (SD) or median (IQR) | N | Mean (SD) or median (IQR) | N | Mean (SD) or median (IQR) |
| | | | mRNA vaccines | | | | | | | | |
| 1 | Danthu 2021 | AU/mL | 14d | NA | NA | NA | NA | 75 | 4 (1.85–12.2) | 7 | 59 (26.5–216.5) |
| | | | 36d | NA | NA | NA | NA | 75 | 6.6 (2.1–19.0) | 7 | 1082 (735–1662) |
| 2 | Fucci 2022 | ng/mL | 22-32d | NA | NA | NA | NA | 155 | 1116 (307.5–9366) | 77 | 4882623 (1177973–5000000) |
| 3 | Grupper 2021 | AU/mL | 30d | NA | NA | NA | NA | 56 | 2900 (1128–5651) | 95 | 7401 (3687–15471) |
| 4 | Jahn 2021 | AU/mL | HD 17d (15–18) Control 13d (13–13) | NA | NA | NA | NA | 72 | 366.5 (89.6–606) | 16 | 800 (520.0–800) |
| 5 | Kolb 2021 | AU/mL | HD 14d (13–15) Control 17d | NA | NA | NA | NA | 32 | 503 (481) | 78 | 1922 (2485) |
| 6 | Labriola 2021 | U/mL | 7d | NA | NA | NA | NA | 24 | 25 (5–250) | 33 | 199 (9–250) |
| 7 | Matsunami 2021 | U/mL | 2-8wk | NA | NA | NA | NA | 78 | 200.5 (116.2–376.5) | 38 | 447 (308.2–1067) |
| 8 | Panizo 2022 | BAU/mL | 15d | 48 | 0 (0–2500) | 14 | (0–114) | 50 | mRNA-1273: 1146 (0–2500) BNT162b2: 381 (0.90–2500) | 16 | mRNA-1273: 641 (0–2500) BNT162b2: 517 (0.90–2500) |
| 9 | Piotrowska | BAU/mL | 14-21d | NA | NA | NA | NA | 35 | 926 (460–1908) | 34 | 2080 (1827–4342) |
| 10 | Piscitani 2021 | IU/mL | 30d | NA | NA | NA | NA | 21 | 492.39 (713.09) | 15 | 1901.20 (287.33) |
| 11 | Schrezenmeier | IU/mL | 4wk | NA | NA | NA | NA | 36 | 74.29 (56.43–86.90) | 44 | 90.91 (77.42–97.05) |
| 12 | Simon 2021 | U/mL | 3wk | NA | NA | NA | NA | 81 | 171 (477.7) | 80 | 2500 (943.5) |
| 13 | Speer 2021a | NA | HD 20d (18–23) Control 19d (19–23) | NA | NA | NA | NA | 124 | 7 (2.8–24.3) | 20 | 134.9 (28.3–283.6) |
| 14 | Speer 2021b | NA | 18-22d | NA | NA | NA | NA | 17 | 6 (1–11) | 46 | 81 (45–150) |
| 15 | Strengert 2021 | RU/mL | 21d | NA | NA | NA | NA | 81 | 272.3 | 34 | 456.8 |
| 16 | Van Praet 2021 (BNT162b2) | AU/mL | 4 or 5w | 322 | 4 | 37 | 3 | 322 | 393 | 37 | 877 |
| | Van Praet 2021 (mRNA-1273) | AU/mL | 4 or 5w | 221 | 4 | 38 | 3 | 221 | 1757 | 38 | 2600 |
| 17 | Zhao 2022 | AU/mL | Dialysis: 105d (range 70–112) Control 117d (range 15–170) | NA | NA | NA | NA | 65 | 168.35 (4.48–1074.29) | 500 | 286.66 (4.72–3556.17) |
| | | | Viral vector vaccines | | | | | | | | |
| 18 | Fu 2022 | U/mL | 4w | 385 | 23.1 (7.3–56.6) | NA | NA | 385 | 602 (307.5–1623) | 66 | 662.5 (391.25–109.25) |
| 19 | Wang 2022 | AU/mL | 4-6w | NA | NA | NA | NA | 204 | 138 (138–140) | 34 | 924 (580.6–1741.5) |
| | Inactivated vaccines | | | | | | | | | | |
| 20 | Bai 2022 | AU/mL | 20d after 1st | NA | NA | NA | NA | 50 | 143.4 (117.8) | 31 | 156.3 (113.8) |
| | | | 3w after 2nd dose | NA | NA | NA | NA | 50 | 180.6 (105.8) | 31 | 186.7 (97.9) |
| 21 | Boongird 2021a | AU/mL | 2w | NA | NA | NA | NA | 60 | 590 (219–1427) | 30 | 1767 (312–7870) |
| 22 | Boongird 2021b | AU/mL | 2w | NA | NA | NA | NA | 30 | 500 (72–2785) | 30 | 1785 (785–3785) |
| 23 | Bruminhent 2022 | BAU/mL | 2w | NA | NA | NA | NA | 31 | 85.3 (33–412.1) | 16 | 250.9 (90.9–612.2) |
| 24 | Dheir 2022 | AU/mL | 28d | NA | NA | NA | NA | 50 | 27.4 (7.8–161.5) | 41 | 74.9 (24.6–270.1) |
| 25 | Murt 2021 | AU/mL | 21-28d | NA | NA | NA | NA | 85 | 408.9 (433.5) | 103 | 685.9 (436.9) |

*(Continued)*

**Table 2.** (Continued)

| No | Author | Unit of IgG level | Time to measurement after vaccination | Baseline data | | | | Follow up data | | | |
|---|---|---|---|---|---|---|---|---|---|---|---|
| | | | | HD group | | Control group | | HD group | | Control group | |
| | | | | N | Mean (SD) or median (IQR) | N | Mean (SD) or median (IQR) | N | Mean (SD) or median (IQR) | N | Mean (SD) or median (IQR) |
| colspan | | | | | | mRNA or viral vector vaccines | | | | | |
| 26 | Lesny 2021 | AU/mL | 2w after 1st dose | 23 | 0.0 (0.0–0.8) | NA | NA | 23 | 1.6 (0–14.5) | 14 | 73.1 (16.1–1324.5) |
| 27 | Kim 2022 | AU/mL | 2m | NA | NA | NA | NA | 100 | 82.1 (34.5–176.6) | 100 | 197.1 (124–346) |
| 28 | Park 2022 | U/mL | 7d | 25 | 0.4 (0) | 55 | 0.4 (0) | 25 | 523.9 (672.9) | 55 | 1192 (881.7) |
| 29 | Tillmann 2021 | AU/mL | 4-5w | NA | NA | NA | NA | 95 | 78 (35) | 60 | 92 (20) |
| colspan | | | | | | mRNA or viral vector or heterologous vaccines | | | | | |
| 30 | Haase 2022 | BAU/mL | 6w | NA | BNT/BNT 0 (0.0–0.3) ChAd/ChAd 0.1 (0.0–0.3) ChAd/BNT 0 (0–0.4) | NA | NA | 100 | BNT/BNT 361 (120–936) ChAd/ChAd 100 (41–346) ChAd/BNT 1744 (276–2840) | 24 | 650 (217–1402) |

Scharpe et al. reported a lower seroconversion rate, but with an insignificant difference, in hemodialysis patients compared to healthy controls, indicating a similar immune response to healthy subjects. In addition, the seroconversion rate is independently related to the baseline seroprotection rate. It is detailed that the baseline seroprotective rate is affected by the frequencies of past immunizations and higher ferritin levels. This study, however, is underpowered to detect a significant difference in immune responses between healthy subjects and hemodialysis patients, with a post hoc power analysis finding that indicated an unrealistically large number of patients would be necessary to achieve an 80% power [60].

Three studies reported an insignificantly higher seroconversion rate in patients with ESRD undergoing hemodialysis than in healthy individuals [26, 50, 58]. However, all three studies are also underpowered (each weighted at 4.6%, 2.6%, and 0.3%) to affect the pooled estimates due to the small number of participants. In addition, one study by Hodges in 1979 still utilized a bivalent split-virus vaccine containing A/New Jersey/76 and A/Victoria/75 instead of a trivalent influenza vaccine [26].

Six studies with considerable heterogeneity were analyzed to generate pooled estimates of the seroprotection rate after H3N2 vaccination in patients with ESRD undergoing hemodialysis. Our study demonstrated a significant decrease of 16% in seroprotection rate in hemodialysis patients compared to healthy subjects. Four of the six studies reported a significantly lower seroprotection rate in patients with ESRD undergoing hemodialysis compared to healthy

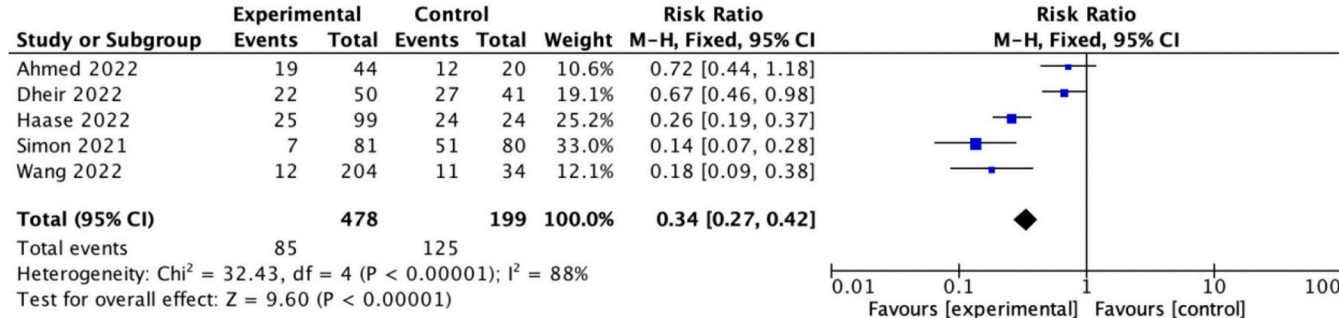

**Fig 5. Forest plot of studies reporting adverse events after COVID-19 vaccination in patients with ESRD undergoing hemodialysis.**

subjects. Furthermore, Eiselt et al. also found a lower seroprotection rate in patients with ESRD undergoing hemodialysis, although the difference was not statistically significant. Nevertheless, Scharpé et al. reported a slightly higher seroprotection rate in hemodialysis patients. Similar to the response to H1N1 influenza vaccination, the higher seroprotection rate might be caused by a higher baseline seroprotection rate in hemodialysis patients due to more frequent immunizations the previous year and the impact of recent advancements in dialysis technology and therapeutic drugs [60].

We found only one study by Scharpé in 2009, which evaluated the safety of H3N2 influenza vaccinations as an outcome. In this study, neither hemodialysis patients nor healthy subjects experienced adverse side effects. Compared to healthy controls, the number of mild adverse events was considerably lower in hemodialysis patients. Hemodialysis patients demonstrated fewer local symptoms, fewer generalized myalgia, and fewer headache symptoms [60]. This finding indicates a more potent immune reaction in healthy subjects compared to hemodialysis patients.

## 4.3 COVID-19 vaccine

Since COVID-19 is a novel disease and numerous different vaccine platforms are currently used, studies investigating immune responses after COVID-19 vaccinations in the HD population also utilize various methods and units of measurement and different vaccine platforms and combinations. Of the included 35 studies investigating COVID-19 vaccination in this systematic review, 30 studies provided data on SARS-CoV-2 IgG antibody response following vaccination (Table 2). Most studies demonstrated lower antibody response in HD patients compared to healthy controls after COVID-19 vaccination, except for one study (i.e., Panizo 2022).

This finding suggests that dialysis patients have a poorer overall antibody response than healthy subjects. As a result, dialysis patients are less likely to be able to neutralize the SARS-CoV-2 virus even after two homologous vaccine doses, no matter the vaccine platform. Thus, vulnerable populations such as hemodialysis patients are more susceptible to infection and severe disease progression [61]. Meanwhile, an interesting finding by Haase et al. 2022 demonstrated higher spike IgG levels in HD patients receiving heterologous vaccination with ChAd/BNT compared to HD patients receiving homologous vaccination with BNT/BNT, ChAd/ChAd, and healthy controls. However, the study did not differentiate the spike IgG levels between different vaccine platforms combinations in the control group. With these findings, a prompt consideration for vaccination dose or schedule adjustment and the administration of heterologous vaccines in ESRD patients on maintenance hemodialysis should be made as done with different vaccines in the past [62].

Meanwhile, a study by Panizo et al. revealed the opposite result. This study demonstrated a higher median anti-RBD IgG level among HD patients (1146 [0–2500] BAU/mL) compared to controls (641 [0–2500] BAU/mL) 15 days after completion of the vaccination schedule with the mRNA-1273 vaccine. This finding might be caused by the larger proportion of seropositive HD patients (12.5%) compared to controls (7%) before vaccination. The participants who were seropositive at baseline might have had a recent COVID-19 infection before vaccination. However, antibody measurement three months after the vaccination showed a waning of antibody levels and a reversal between the two groups (HD: 388 [0–2500] BAU/mL vs. Control: 477 [5.9–2500] BAU/mL). The more pronounced decline in HD patients suggests accelerated kinetics of antibody waning in this population [36].

Even though the gold standard to measure the neutralizing capacity of patients' serum antibodies is a plaque reduction neutralization test [63], the anti-SARS-CoV-2 S antibody has been

shown to have a high correlation with a direct virus neutralization test and a surrogate neutralization assay [64]. Therefore, the anti-SARS-CoV-2 antibody can be used as a surrogate marker for vaccine-induced immunity.

This review demonstrated that, generally, patients with ESRD undergoing hemodialysis have a blunted early serological response to SARS-CoV-2 vaccination. The dynamics of humoral immune response to different SARS-CoV-2 vaccines in this population may be affected by several factors, such as the use of immunosuppressive medications, dialysis vintage, and previous history of COVID-19 vaccination. A multivariate analysis from a prospective cohort study conducted by Van Praet et al. revealed that COVID-19 experience, immunosuppressive drugs use, and dialysis vintage represent independent predictors of humoral immune responses (Van Praet 2021). However, not all included studies in this review provided the data on immunosuppressive drugs and dialysis vintage (extracted data available in https://osf.io/es2ma/?view_only=87b0e57246704617aa094219a60ba73b).

Pooled estimates of the adverse events rate after COVID-19 vaccination were derived from five studies with substantial heterogeneity ($I^2 = 88\%$). Four studies showed a significantly lower number of adverse events in ESRD patients undergoing hemodialysis compared to healthy controls. The pooled estimates in our study demonstrated a 66% lower percentage of adverse events rate in ESRD patients undergoing hemodialysis. This result represents a more potent and noticeable immune reaction in cellular and humoral arms in healthy individuals. The correlation of adverse events with the amount of immunosuppression and whether the number of AEs can indirectly predict response to vaccination are potential research topics to be explored in the future. Further studies are needed to determine the potential causal relationship between adverse events and immune response in patients with ESRD on hemodialysis.

To our knowledge, this review is the first to investigate vaccination against respiratory diseases in ESRD patients undergoing hemodialysis. The overall quality of evidence for seroconversion and seroprotection rate after both H1N1 and H3N2 vaccination and the adverse events rates in COVID-19 vaccination was assessed using the GRADE framework (S1 Table).

There are several limitations of our study. In the absence of RCT data, serological conversion represents the most appropriate surrogate for efficacy despite not being a true measure. Antibody titer data were extracted. However, due to heterogeneous measurement methods, pooled analyses could not be performed. Secondly, due to a lack of available data, our discussion on vaccine safety was limited. In addition, data on immunosuppressive medications, the onset of dialysis, the glomerular filtration rate, and other predictors potentially influencing the immunogenicity outcomes were also inadequate.

## 5. Conclusions

Our systematic review demonstrates evidence of lower seroconversion and seroprotection rates after vaccinations against viral respiratory diseases in ESRD patients undergoing hemodialysis. We consistently found a lower incidence of minor adverse events and no reported serious adverse events in hemodialysis patients after vaccination. Considering that hemodialysis patients are more susceptible to infection and severe disease progression, a weakened yet substantial serological response can be considered adequate for the recommendation of vaccination against respiratory diseases vaccination in this population. Vaccination dose, schedule, or strategy adjustments should be considered in ESRD patients undergoing hemodialysis.

## Supporting information

**S1 Checklist. PRISMA 2009 checklist.**
(PDF)

**S1 Protocol. Protocol of systematic review.**
(PDF)

**S1 Appendix. Database searching strategy.**
(PDF)

**S2 Appendix. Funnel plots of studies included in the meta-analyses.**
(PDF)

**S1 Table. Grading of Recommendations, Assessment, Development, and Evaluation (GRADE) criteria for studies included in the meta-analyses.**
(PDF)

## Acknowledgments

Authors express gratitude to the staff of Klinik Bahasa in the Office of Research and Publication, Faculty of Medicine, Public Health and Nursing, Universitas Gadjah Mada, Yogyakarta, Indonesia for the English language and grammar editing of the manuscript.

## Author Contributions

**Conceptualization:** Metalia Puspitasari, Prenali D. Sattwika.

**Data curation:** Metalia Puspitasari, Prenali D. Sattwika, Dzerlina S. Rahari, Wynne Wijaya, Auliana R. P. Hidayat.

**Formal analysis:** Metalia Puspitasari, Prenali D. Sattwika, Dzerlina S. Rahari, Wynne Wijaya.

**Funding acquisition:** Metalia Puspitasari.

**Investigation:** Metalia Puspitasari, Prenali D. Sattwika, Dzerlina S. Rahari, Wynne Wijaya, Auliana R. P. Hidayat.

**Methodology:** Metalia Puspitasari, Prenali D. Sattwika, Dzerlina S. Rahari, Wynne Wijaya, Auliana R. P. Hidayat.

**Project administration:** Metalia Puspitasari, Dzerlina S. Rahari, Wynne Wijaya, Auliana R. P. Hidayat.

**Resources:** Metalia Puspitasari, Prenali D. Sattwika, Dzerlina S. Rahari, Wynne Wijaya, Auliana R. P. Hidayat.

**Software:** Metalia Puspitasari, Prenali D. Sattwika, Dzerlina S. Rahari, Wynne Wijaya, Auliana R. P. Hidayat.

**Supervision:** Nyoman Kertia, Bambang Purwanto, Jarir At Thobari.

**Validation:** Metalia Puspitasari, Prenali D. Sattwika, Nyoman Kertia, Bambang Purwanto, Jarir At Thobari.

**Writing – original draft:** Metalia Puspitasari, Dzerlina S. Rahari, Wynne Wijaya, Auliana R. P. Hidayat.

**Writing – review & editing:** Metalia Puspitasari, Prenali D. Sattwika, Dzerlina S. Rahari, Wynne Wijaya, Auliana R. P. Hidayat.

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
