## [Decision Letter · Decision Letter 0]

19 Oct 2022

PONE-D-22-27621Outcomes of vaccinations against respiratory diseases in patients with end-stage renal disease undergoing hemodialysis: a systematic reviewPLOS ONE

Dear Dr. Puspitasari,

Thank you for submitting your manuscript to PLOS ONE. After careful consideration, we feel that it has merit but does not fully meet PLOS ONE’s publication criteria as it currently stands. Therefore, we invite you to submit a revised version of the manuscript that addresses the points raised during the review process. In PLOS ONE, the methodology is important. Please revise this point carefully. 

We look forward to receiving your revised manuscript.

Kind regards,

Etsuro Ito

Academic Editor

PLOS ONE

Journal Requirements:

3. Please identify your study as "systematic review and meta-analysis" in the title.

5. Please upload a copy of Supporting Information Table. S2 Table which you refer to in your text on page 4.

Reviewers' comments:

Reviewer's Responses to Questions

**Comments to the Author**

1. Is the manuscript technically sound, and do the data support the conclusions?

Reviewer #1: Partly

Reviewer #2: Yes

2. Has the statistical analysis been performed appropriately and rigorously? 

Reviewer #1: Yes

Reviewer #2: Yes

3. Have the authors made all data underlying the findings in their manuscript fully available?

Reviewer #1: No

Reviewer #2: Yes

4. Is the manuscript presented in an intelligible fashion and written in standard English?

Reviewer #1: Yes

Reviewer #2: Yes

5. Review Comments to the Author

Reviewer #1: Thank you for submitting your research, I enjoyed reading your review. Please find below some methodological comments, which I hope will support you to further improving your work:

- Title: you should mention that your work is a systematic review AND a meta-analysis;

- Please add the"highlights" section per bullet points with your core findings;

- Open Science: I recommend to add your the full tables of your data extraction and any additional material as publicly available (e.g. the Open Science Framework (OSF) portal is a useful resource and then you can add a link to your OSF page in your manuscript)

- Introduction: this section need major improvements to better highlight the evidence gap and the relevance of your research.

- Methods:

a) it is not clear if you refined your search strategy with the support of a libriarian and/or an expert in literature searches;

b) you should declare the start date also of your search (at the moment only the end date is stated);

c) your search strategy MeSH terms and databases is not well defined: did you search only in Pubmed/Medline? why not Scopus, WoS, CINAHL, Cochrane library, ProQuest, Science Direct or other dababases (maybe also including grey literature from Google scholar)? Your search strategy in its current state is methodologically questionable.

d) Have you considered to perform a Cohen's K to quantify reviewers' agreement? This would strenghten your methodology;

e) selection criteria would be better namend as "eligibility criteria";

f) I2 values and low, moderate or high level of heterogeneity would be better describes by a percentage range. To my knowledge and according to the Cochrane standards: heterogeneity is not important if I2 ranges from 0% to 40%, mod-

erate from 30% to 60%, substantial from 50% to 90% and considerable from 75% to 100%;

g) I understand there are only few studies included but I strongly recommend to consider a funnel plot as well to check a publication bias;

- Results:

a) I would recommend to shape your tables according to the APA style;

b) the risk of bias table needs to comply with the standards (e.g. look at https://guides.library.cornell.edu/evidence-synthesis/bias);

c) I'm surprised to see studies from 1988 and 1985 included: I supposed the standards of care, vaccination and measurements of outcomes were different at that time. This is potentially a flaw. Also, I would specify in your methodology the timespan of your search and why. For example when the last systematic review has been performed about this topic? Your should then start your search from there.

Reviewer #2: The study is well done and the statistical methods appear robust. There are some minor typos and grammatical errors that should be fixed prior to publication. The data tables are spread across several pages and would benefit from additional formatting to improve readability and interpretation.

The authors use serological conversion as a surrogate for efficacy. This is not a true measure of efficacy, but is the most appropriate surrogate in the absence of RCT data. It may be important to make a statement on the rationale for serological conversion as a measure of efficacy. Due to a lack of available data, the authors' discussion on vaccine safety was also limited. Unfortunately, RCT on these vaccines have not been conducted in this population group, and safety data is therefore limited. The author's discussion of adverse events data is the most suitable replacement for true efficacy data, but it would be worthwhile to explain why this measure needed to be used.

Furthermore, some causes of ESRD may require immunosuppressive medications that would further distinguish these patients from other patients with ESRD. If these data are available, mentioning them in the manuscript would be worthwhile as these patients may have drastically different seroconversion rates compared to other ESRD patients. If these studies include information on patient time on dialysis or patient GFR, these data would also be worth including in the manuscript.

Overall, the manuscript is well put together and the methods appear robust. With some additional considerations as mentioned above, this will be a very high quality manuscript worth of publication.

6. PLOS authors have the option to publish the peer review history of their article (what does this mean?). If published, this will include your full peer review and any attached files.

Reviewer #1: No

Reviewer #2: **Yes: **Nicolas F Moreno

---

## [Author Response · Author response to Decision Letter 0]

3 Jan 2023

Dear Reviewers,

We would like to appreciate your constructive feedback. We are providing point-by-point responses as follows:

Journal Requirements:

We have double-checked that this manuscript meets PLOS ONE’s style requirements.

The grammar and spelling of this manuscript have been proofread by a native speaker from the language clinic provided by our institution, thank you.

3. Please identify your study as "systematic review and meta-analysis" in the title.

Thank you, we have revised the title.

4. PLOS requires an ORCID iD for the corresponding author in Editorial Manager on papers submitted after December 6th, 2016. Please ensure that you have an ORCID iD and that it is validated in Editorial Manager.

We have authorized the ORCID ID (https://orcid.org/0000-0002-8884-4579) of the corresponding author.

5. Please upload a copy of the Supporting Information Table. S2 Table which you refer to in your text on page 4.

The supplementary files have been completed, including S2 Appendix of database searching strategy and S4 Appendix of funnel plots. 

Reviewers' Comments:

Reviewer #1:

- Title: you should mention that your work is a systematic review AND a meta-analysis; 

Thank you, we have revised the title.

- Please add the"highlights" section per bullet points with your core findings; 

We have added the highlights section. Please refer to page 3, thank you.

- Open Science: I recommend to add your the full tables of your data extraction and any additional material as publicly available (e.g. the Open Science Framework (OSF) portal is a useful resource and then you can add a link to your OSF page in your manuscript) 

Additional materials can be accessed via this OSF link: https://osf.io/es2ma/?view_only=87b0e57246704617aa094219a60ba73b

Thank you for the suggestion.

- Introduction: this section need major improvements to better highlight the evidence gap and the relevance of your research. 

We have revised the introduction section (highlighted in yellow) to provide current knowledge and emphasize the research gap to be covered by this systematic review. Thank you.

- Methods: 

a) it is not clear if you refined your search strategy with the support of a libriarian and/or an expert in literature searches; 

We have discussed and refined our search strategy with an expert in literature searches from our institution, thank you.

b) you should declare the start date also of your search (at the moment only the end date is stated); 

We have specified the date of searching, from inception until 20 October 2022.

c) your search strategy MeSH terms and databases is not well defined: did you search only in Pubmed/Medline? why not Scopus, WoS, CINAHL, Cochrane library, ProQuest, Science Direct or other dababases (maybe also including grey literature from Google scholar)? Your search strategy in its current state is methodologically questionable. 

We have extended our search in several databases subject to access availability, for instance: Scopus, Cochrane Library, Google Scholar, ScienceDirect, and ProQuest. Our apologies due to limited access to certain databases (WoS and CINAHL) we could not perform searching on those databases. Thank you for this valuable feedback as we managed to identify additional relevant studies to be included in our review. We added a total of 28 studies (highlighted in yellow in Table 1).

d) Have you considered to perform a Cohen's K to quantify reviewers' agreement? This would strenghten your methodology; 

Thank you for the suggestion, however, we could not proceed to Cohen’s K measurement in this time frame. Our action plan is to attend relevant training; therefore, this will improve our review in the future.

e) selection criteria would be better named as "eligibility criteria”; 

We have replaced the wording to be eligibility criteria, thank you.

f) I2 values and low, moderate or high level of heterogeneity would be better describes by a percentage range. To my knowledge and according to the Cochrane standards: heterogeneity is not important if I2 ranges from 0% to 40%, moderate from 30% to 60%, substantial from 50% to 90% and considerable from 75% to 100%; 

Thank you for this insight, we have improved the description for heterogeneity.

g) I understand there are only few studies included but I strongly recommend to consider a funnel plot as well to check a publication bias;

We have added funnel plots as S4 appendix in the supplementary materials, thank you.

- Results: 

a) I would recommend to shape your tables according to the APA style; 

We have edited the layout of our tables according to the APA style.

b) the risk of bias table needs to comply with the standards (e.g. look at https://guides.library.cornell.edu/evidence-synthesis/bias); 

Thank you, we have improved the risk of bias results (Fig 2) to comply with the above-mentioned standards.

c) I'm surprised to see studies from 1988 and 1985 included: I supposed the standards of care, vaccination and measurements of outcomes were different at that time. This is potentially a flaw. Also, I would specify in your methodology the timespan of your search and why. For example when the last systematic review has been performed about this topic? Your should then start your search from there.

Thank you for the input. We decided to search from inception to get a general idea of studies conducted in ESRD patients undergoing hemodialysis. We are aware that the 80s studies do not reflect current clinical conditions and should be cautiously interpreted.

Reviewer #2: The study is well done and the statistical methods appear robust. There are some minor typos and grammatical errors that should be fixed prior to publication. 

Thank you for your feedback, we have double-checked to correct the typos and grammatical errors.

The data tables are spread across several pages and would benefit from additional formatting to improve readability and interpretation. 

We have reformatted our tables according to the APA style, thank you.

The authors use serological conversion as a surrogate for efficacy. This is not a true measure of efficacy, but is the most appropriate surrogate in the absence of RCT data. It may be important to make a statement on the rationale for serological conversion as a measure of efficacy. 

Thank you for adding this point of view. We have provided the rationale for choosing serological conversion as this is the parameter that could be analysed across the studies with different techniques of antibody measurement (highlighted in yellow in Section 4.1).

Due to a lack of available data, the authors' discussion on vaccine safety was also limited. Unfortunately, RCT on these vaccines have not been conducted in this population group, and safety data is therefore limited. 

We have provided additional studies to improve our data. However, only limited studies assess the safety of vaccinations. Therefore, we have also mentioned this aspect as one of the limitations. Thank you.

The author's discussion of adverse events data is the most suitable replacement for true efficacy data, but it would be worthwhile to explain why this measure needed to be used. 

Thank you for your suggestion. We added more discussion to address the efficacy data (highlighted in yellow).

Furthermore, some causes of ESRD may require immunosuppressive medications that would further distinguish these patients from other patients with ESRD. If these data are available, mentioning them in the manuscript would be worthwhile as these patients may have drastically different seroconversion rates compared to other ESRD patients. If these studies include information on patient time on dialysis or patient GFR, these data would also be worth including in the manuscript. 

If available, the data of immunosuppressive medication, the onset of dialysis, and the GFR are provided in OSF. Additional discussion has been added as well. Thank you.

Overall, the manuscript is well put together and the methods appear robust. With some additional considerations as mentioned above, this will be a very high quality manuscript worth of publication.

We appreciate your valuable insight and have made substantial revisions, thank you. 

Any feedback following our revision is most welcomed.

Thank you in advance.

Best wishes,

Metalia Puspitasari

---

## [Decision Letter · Decision Letter 1]

17 Jan 2023

Outcomes of vaccinations against respiratory diseases in patients with end-stage renal disease undergoing hemodialysis: a systematic review and meta-analysis

PONE-D-22-27621R1

Dear Dr. Puspitasari,

We’re pleased to inform you that your manuscript has been judged scientifically suitable for publication and will be formally accepted for publication once it meets all outstanding technical requirements.

Kind regards,

Etsuro Ito

Academic Editor

PLOS ONE

Reviewers' comments:

Reviewer's Responses to Questions

**Comments to the Author**

1. If the authors have adequately addressed your comments raised in a previous round of review and you feel that this manuscript is now acceptable for publication, you may indicate that here to bypass the “Comments to the Author” section, enter your conflict of interest statement in the “Confidential to Editor” section, and submit your "Accept" recommendation.

Reviewer #1: All comments have been addressed

Reviewer #2: All comments have been addressed

2. Is the manuscript technically sound, and do the data support the conclusions?

Reviewer #1: Yes

Reviewer #2: Yes

3. Has the statistical analysis been performed appropriately and rigorously? 

Reviewer #1: Yes

Reviewer #2: Yes

4. Have the authors made all data underlying the findings in their manuscript fully available?

Reviewer #1: Yes

Reviewer #2: Yes

5. Is the manuscript presented in an intelligible fashion and written in standard English?

Reviewer #1: Yes

Reviewer #2: Yes

6. Review Comments to the Author

Reviewer #1: Thank you for addressing properly my comments, your manuscript is now improved from my perspective, I have no further comments.

Reviewer #2: Thank you for addressing the feedback and reviewer comments. Your manuscript is high quality in the present state and provides valuable insight into special considerations for vaccination of hemodialysis patients against respiratory diseases.

7. PLOS authors have the option to publish the peer review history of their article (what does this mean?). If published, this will include your full peer review and any attached files.

Reviewer #1: No

Reviewer #2: **Yes: **Nicolas F Moreno

---

## [Editor Report · Acceptance letter]

31 Jan 2023

PONE-D-22-27621R1 

Outcomes of vaccinations against respiratory diseases in patients with end-stage renal disease undergoing hemodialysis: a systematic review and meta-analysis 

Dear Dr. Puspitasari:

I'm pleased to inform you that your manuscript has been deemed suitable for publication in PLOS ONE. Congratulations! Your manuscript is now with our production department. 

Kind regards, 

on behalf of

Prof. Etsuro Ito 

Academic Editor

PLOS ONE